physiology, behaviour, biomechanics

muscle fatigue, jaws, muscle histochemistry, myosin isoform, bite force, *Elgaria*

**Author for correspondence:**
A. Kristopher Lappin
e-mail: aklappin@cpp.edu

# Fatigue resistant jaw muscles facilitate long-lasting courtship behaviour in the southern alligator lizard (*Elgaria multicarinata*)

Allyn Nguyen[1], Jordan P. Balaban[2], Emanuel Azizi[2], Robert J. Talmadge[1] and A. Kristopher Lappin[1]

[1]Biological Sciences Department, California State Polytechnic University, Pomona, CA 91768, USA
[2]Department of Ecology and Evolutionary Biology, University of California, Irvine, CA 92697, USA

 AN, 0000-0003-0006-3565; AKL, 0000-0002-5386-0069

The southern alligator lizard (*Elgaria multicarinata*) exhibits a courtship behaviour during which the male firmly grips the female's head in his jaws for many hours at a time. This extreme behaviour counters the conventional wisdom that reptilian muscle is incapable of powering high-endurance behaviours. We conducted *in situ* experiments in which the jaw-adductor muscles of lizards were stimulated directly while bite force was measured simultaneously. Fatigue tests were performed by stimulating the muscles with a series of tetanic trains. Our results show that a substantial sustained force gradually develops during the fatigue test. This sustained force persists after peak tetanic forces have declined to a fraction of their initial magnitude. The observed sustained force during *in situ* fatigue tests is consistent with the courtship behaviour of these lizards and probably reflects physiological specialization. The results of molecular analysis reveal that the jaw muscles contain masticatory and tonic myosin fibres. We propose that the presence of tonic fibres may explain the unusual sustained force properties during mate-holding behaviour. The characterization of muscle properties that facilitate extreme performance during specialized behaviours may reveal general mechanisms of muscle function, especially when done in light of convergently evolved systems exhibiting similar performance characteristics.

## 1. Introduction

Many lizards exhibit mate-holding behaviour (e.g. skinks, lacertids, iguanians, etc.) in which the male uses his jaws to grasp the female's neck, typically by a fold of skin, or at another part of the body [1–4]. However, few male lizards perform mate-holding behaviour by grasping the entire head of the female for extended periods, as is the case with the southern alligator lizard (*Elgaria multicarinata*). During courtship in this species of anguid lizard, the male firmly grips the female's head in his jaws for many hours at a time. A.K.L. (personal observation, 27–29 November 2004) observed a large male *E. multicarinata* continuously hold a female for at least 40 h. Further, a citizen science project at the Natural History Museum of Los Angeles County [5] has produced numerous accounts of the behaviour by *E. multicarinata*, as well as by the closely related northern alligator lizard (*E. coerulea*). Some records document lizard pairs exhibiting mate-holding behaviour for over a day and even up to two days (i.e. 25 and 49 h). Although there is a lack of published accounts of mate-holding behaviour by male *E. multicarinata*, it was described in *E. coerulea* (*Gerrhonotus coeruleus*) by Vestal [6], Svihla [7] and Vitt [8]. Online searches reveal images of limbless anguids (e.g. *Pseudopus apodus* and *Anguis fragilis*)

showing mate-holding behaviour identical to that documented in *E. multicarinata* and *E. coerulea*, suggesting that the behaviour may be typical of the Anguidae. It is unknown whether mate-holding behaviour by anguid lizards represents mate guarding, a demonstration of strength or endurance to the female, and/or is associated with other aspects of reproductive biology.

Mate holding for many hours at a time by male *E. multicarinata* constitutes an extreme behaviour that indicates physiological specialization for fatigue resistance that is atypical of vertebrate skeletal muscle. Quantification of muscle contractile properties can aid in understanding the physiological role of specific muscles in extreme behaviours (e.g. [9–12]). For example, when a muscle-powered behaviour exhibits extreme endurance, fatigue tests that quantify the decline in muscle force generation and characteristics of relaxation with repeated tetanic contractions can be illuminating (e.g. [13]). An example of the ability of muscle to sustain appreciable force for long periods is found in the amplectic muscles of males of some sexually dimorphic frogs [13]. These amplectic muscles exhibit a sustained force profile during *in situ* experiments (i.e. constant force maintained between tetanic contractions), which presumably facilitates hours-long amplexus [13–15].

Muscle fibre types and their physiological characteristics contribute to muscle performance in terms of force production, shortening velocity and fatiguability. Most of the published literature recognizes three general types of twitch fibres: slow oxidative (SO/type I), fast glycolytic (FG/type IIB or IIx) and fast oxidative glycolytic (FOG/type IIA). This nomenclature is used universally in the literature to describe most muscle fibres, but there are fibres in some systems that exhibit characteristics that differ from the three generally recognized types of twitch fibres.

Tonic muscle fibres differ from the three types of twitch fibres in key aspects, and they may play a role in the extreme courtship behaviour of *E. multicarinata*. Found in some muscles (e.g. jaw, limb, trunk), tonic fibres are a type of slow fibre [16–21]. Tonic fibres have unique contractile properties and do not contract with a twitch, but rather exhibit a prolonged, fatigue-resistant contraction that is similar to that of mammalian smooth muscle [22]. Slow-twitch and slow–tonic fibres both have been found in turtle hindlimb and neck muscles, and the two types have been identified as distinct from one another [23]. The primary role of tonic fibres is to facilitate fatigue-resistant behaviours involving stabilization and maintenance of postures in a variety of vertebrates [23–27], though they also may be involved in very slow locomotor performance as observed in chameleons [28] (but see [29]).

Despite observational data indicating that the jaw-adductor muscles of *E. multicarinata* exhibit distinct characteristics that facilitate the production of long-lasting sustained force, the contractile properties and fibre type composition of this muscle complex are unknown. An investigation of the contractile properties of *E. multicarinata* jaw-adductor muscles, particularly in terms of fatigue, may reveal how the musculature behaves during mate-holding behaviour. Characterization of the biochemical properties of the jaw-adductor muscles may provide clues as to the underlying mechanism(s) that contribute to the muscles' extreme performance.

To quantify the contractile properties of the jaw-adductor muscles of *E. multicarinata*, we conducted *in situ* experiments in which the jaw muscles of lizards were stimulated directly while bite force was measured simultaneously with a double-cantilever beam force transducer. To investigate the fibre type composition of the jaw muscles we performed SDS–PAGE gels, western blots, and muscle immunohistochemistry. We hypothesize that the jaw-adductor muscle complex, under controlled experimental conditions, will (i) show evidence of high resistance to fatigue and (ii) contain a slow myosin heavy chain (MHC)-based fibre type (i.e. tonic) that could contribute to sustained force characteristics apparent during mate-holding behaviour in this species.

## 2. Material and methods

Southern alligator lizards (*E. multicarinata*; 4 males, 4 females) were wild-caught by hand in Claremont, California. Lizards were kept in terraria under controlled temperature and light conditions ($24 \pm 2°C$; 12 h : 12 h light : dark) and provided with cover objects for hiding. They were fed crickets dusted with a calcium and vitamin supplement every other day and had access to water *ad libitum*. Lizards were maintained in captivity for up to a year prior to being used in experiments. *Elgaria multicarinata* is easy to maintain in captivity, and all specimens remained in excellent condition from the time of collection to the time they were used in experiments. All specimens were adults when collected and did not vary greatly in body or head size (mean ± s.e.m: full sample—snout–vent length: $128.1 \pm 3.22$ mm; body mass: $44.1 \pm 3.70$ g; head length: $24.5 \pm 0.57$ mm; head width: $18.2 \pm 0.59$ mm; head depth: $14.4 \pm 0.55$ mm; males—snout–vent length: $124.9 \pm 4.62$ mm; body mass: $37.0 \pm 3.95$ g; head length: $25.1 \pm 1.03$ mm; head width: $18.6 \pm 1.08$ mm; head depth: $14.8 \pm 0.88$ mm; females: snout–vent length: $131.4 \pm 4.49$ mm; body mass: $51.1 \pm 3.95$ g; head length: $24.0 \pm 0.52$ mm; head width: $17.8 \pm 0.60$ mm; head depth: $14.0 \pm 0.74$ mm; see [30] for head measurement methods). At the conclusion of the experiments, a midline laparotomy was performed to reveal the reproductive organs (oviducts and testes) to confirm sex. All procedures were approved by the Institutional Animal Care and Use Committees at the University of California, Irvine, and at California State Polytechnic University, Pomona.

### (a) Measurement of muscle contractile properties

Lizards were anesthetized with 5% isoflurane and double-pithed. The integument overlying the lateral temporal fenestration on each side of the head was removed to expose the jaw-adductor muscles (adductor mandibulae complex). The muscles were kept moist during experiments by frequent irrigation with Ringer's solution (NaCl 16.95 g, KCl 0.60 g, imidazole 2.72 g, CaCl₂ 0.55 g, glucose 3.96 g per 2 l of distilled water) [31].

We measured bite force *in situ* during stimulation of the jaw-adductor complex (4 males; 4 females) at $24 \pm 1°C$ in a temperature-controlled room. The specimen was positioned on a platform, and the tips of the jaws were manually placed over the bars of a custom-built stainless steel, double-cantilever beam force transducer that was fixed in place with a clamp adjacent to the platform. The transducer was equipped with linear strain gauges (Omega, SGD-3/350-LY11) wired in a half-bridge configuration. We glued a leather strip (6 × 18 mm) to the end of the outer surface of each 20 mm wide bar so that once the lizard's teeth engaged the transducer, its jaws would remain in a constant position to avoid confounding changes in bite outlever during the trial [32]. The exposed jaw-adductor complex was implanted bilaterally with custom-made gold electrodes with 3.5 mm long two-pronged tips separated by 2 mm (Amphenol SINE systems, USA) into the centre of the muscle complex to ensure that as much of the muscle tissue was stimulated as possible during the trials. Preliminary trials using two pairs of

electrodes in each muscle did not result in higher forces indicating that a single pair of electrodes per side was effective in stimulating the muscle complex. Muscles were stimulated using a GRASS S48 stimulator, and the voltage output from the strain gauges was amplified (Model 2120; Vishay Precision Group; Wendell, NC) and recorded at 1000 Hz using data acquisition software (IgorPro; WaveMetrics; Lake Oswego, OR). At the conclusion of each experiment (i.e. after fatigue test), the lizard's thoracic girdle region was exposed to confirm that the heart was still beating to ensure that the jaw muscles were supplied with blood for the duration of the experiment.

We began each experimental session with a series of individual twitch contractions (0.2 ms stimulation duration) induced at increasing voltages to establish the supramaximal voltage required to attain maximum twitch contraction force by the muscle complex for each individual. We did not attempt to identify the gape angle at which maximum force would be achieved, as our focus was primarily on the fatigue characteristics of the jaw-adductor complex. Once the supramaximal voltage was determined (10–15 V with a set resistance of 250 ohms giving 0.04–0.06 amps), we then delivered trains of 0.2 ms pulses at increasing pulse frequencies (5, 10, 15, 20, 25 Hz) to determine the pulse rate at which tetanic fusion occurs. We allowed the muscles to rest for two minutes between each stimulus train to minimize fatigue, as preliminary experiments indicated that this rest duration was sufficient for the muscle complex to recover. Time to peak tension (TPT) was measured from the onset of the stimulus to the peak of twitch tension. Half-relaxation time (½RT) was measured from the peak of twitch tension to the point at which the force was half of the peak value. Measurements analogous to twitch TPT and ½RT were made for tetanic contractions at 25 Hz pulse frequency. We used 25 Hz for the stimulation frequency because at this frequency tetanic fusion occurred in all muscle preparations tested. We used the maximum force produced by the muscle complex at each stimulation frequency to generate a force–frequency curve for each specimen.

To quantify fatiguability, the muscles were stimulated repeatedly with 150 ms trains (0.2 ms pulses at 60 Hz) with one train occurring every 3 s. We used a stimulation frequency of 60 Hz for the fatigue tests to ensure that all of the fibres in the muscle complex were being fully activated. The fatigue test was run for a total of 5 min. During the fatigue test for all specimens, we observed that the force generated by the jaw-adductor muscle complex between tetanic trains gradually rose and then plateaued. Therefore, for each fatigue test, we identified the peak tetanic force and the minimum force between tetani (i.e. sustained force) within each 30-s interval, beginning at time = 0. We then plotted the peak tetanic forces and the sustained forces to examine the change in each over the course of the fatigue test.

## (b) Myosin heavy chain isoform characterization

The adductor mandibulae externus superficialis jaw muscle and ambiens thigh muscle were bilaterally removed from two freshly euthanized lizards (1 male; 1 female), snap frozen in liquid nitrogen, and stored at −80°C until analysis. We used these samples for immunohistochemistry followed by single-fibre SDS–PAGE gel electrophoresis. One additional pair of jaw muscles was dissected from another male *E. multicarinata* and placed directly into a relaxing solution overnight for subsequent single-fibre extraction and then frozen at −20°C for single-fibre gel electrophoresis.

Skeletal muscle MHC isoforms were separated by electrophoresis as described previously [33]. In brief, myofibrillar protein was isolated by homogenization in RIPA buffer and total protein concentration determined [34]. Protein homogenates were reduced (boiled at 100°C for 5 min) and run on a sodium dodecyl sulfate—polyacrylamide mini gel system (SDS–PAGE) at 150 V for 24 h. Electrophoresis of MHC of the *E. multicarinata*

jaw and thigh muscle samples were run alongside a mouse gastrocnemius sample using 8.5% gel. Protein bands were detected in the polyacrylamide gel using a commercially available silver stain kit (BioRad; Hercules, CA).

We identified MHC isoforms by western blotting using established methods [35]. Following electrophoresis, proteins were transferred to polyvinylchloride membranes, incubated in blocking solution (BSA), and then incubated in primary antibodies overnight at 4°C. Primary antibodies were 2F4c (anti-MHC IIm; 1 : 100), ALD-58 (anti-MHC I; 1 : 100), BF-35 (1 : 10 000), D5 (anti-MHC I; 1 : 10), F3 (anti-MHC IIb; 1 : 10) and Sigma Fast (anti-MHC II; 1 : 1000). Antibody BF-35 was supplied by S. Schiaffino (University of Padova, Padova, Italy). All other primary antibodies were obtained from the Developmental Studies Hybridoma Bank at the University of Iowa, Iowa City, IA. Membranes were washed and incubated with appropriate secondary antibodies (anti-Mouse at 1 : 1000; Sigma and anti-donkey at 1 : 1000; Sigma) conjugated to horse-radish peroxidase for 1 h at room temperature. Antibody binding was visualized using SuperSignal West Pico PLUS Chemiluminescent Substrate (Pierce; Rockford, IL) and an Alpha Innotech FluorChem imaging system. Western blots are reverse images.

Serial cryostat sections taken at −18°C (10 μm) were used for immunohistochemistry [36]. Monoclonal antibodies (MAbs) 2F4c, ALD-58, D5, F3, Sigma Fast, SC-71 (anti-MHC IIa; 1 : 1000), A4.15.19 (anti-MHC IIa; 1 : 10), BF-13 (anti-MHC II; 1 : 10 000), N3.36 (neonatal and adult fast fibres; 1 : 10) and Vector Slow (anti-MHC I; 1 : 100) were used to verify the presence of specific MHC isoforms.

Single-fibre electrophoresis was performed to determine whether two myosin isoforms are present in individual fibres of the jaw muscle, given that two isoforms were apparent in the homogenized muscle gel electrophoresis. For the single-fibre electrophoresis, the remaining jaw-adductor muscles from the immunohistochemistry experiments and the jaw-adductor muscles from a third individual were placed in relaxing solution in −20°C [37] for 24 h for muscle fibre isolation. A total of 80 fibres were isolated from the jaw-adductor muscle among the three individuals (2 males; 1 female) using Thermo Fisher Scientific fine dissecting forceps and a Leica EZ4 stereo microscope. Individual fibres were placed in 20 μl of 1× SDS–PAGE buffer with beta-mercaptoethanol. Samples were stored frozen at −20°C until electrophoresis was performed.

## (c) Statistical analyses

To compare maximum tetanic forces realized at the different stimulation frequencies tested in the force–frequency experiments (5, 10, 15, 20, 25 Hz), we performed one-way ANOVA. Bite force was the dependent variable, and the independent variables were stimulation frequency (fixed effect) and individual (random effect). Given that only males exhibit jaw-mediated mate-holding behaviour, we compared sustained force between the sexes. To do this, we first calculated residuals from maximum sustained force between tetanic trains regressed on maximum tetanic force, to control for variation in the size and force-generating capacity of the jaw-adductor complex among individuals. We then performed a one-way ANOVA with the residuals as the dependent variable and sex (fixed effect) and individual (random effect) as the independent variables. The statistical analysis was performed using JMP v. 13.0.0.

## 3. Results

### (a) Force–frequency relationship

Stimulation of the jaw-adductor muscle complex at a pulse frequency of 5 Hz (0.2 ms pulse duration) produced temporal

summation in all specimens (figure 1a). Increasing the frequency of stimulation to 10 Hz resulted in partial fusion and incomplete tetanus. At 15 Hz, we observed near complete fusion, and stimulation frequencies of 20 and 25 Hz produced complete tetanus of the muscle complex. The TPT for a twitch contraction, measured from the first twitch in the at 5 Hz test, was $63.1 \pm 3.7$ ms, and the ½RT was $266 \pm 47.8$ ms. For a tetanic contraction (i.e. 25 Hz), TPT = $762.5 \pm 48.8$ ms, and ½RT = $1787.2 \pm 145.7$ ms.

To determine the relationship between the frequency of stimulation and the maximum force produced by the jaw-adductor muscle complex, we plotted a force–frequency curve based on the averages among the individuals (figure 1b). The maximum force produced by the muscle complex at the different stimulation frequencies (5, 10, 15, 20, 25 Hz) was not found to differ statistically (figure 1b; $F = 50.9246$, d.f. = 4, $p = 0.0881$).

## (b) Fatigue profile

Stimulation of the muscle complex at supramaximal voltage elicited a series of tetanic contractions that exhibited a similar pattern of force development and fatigue for all specimens. During the fatigue test, as expected, there was a decrease in the peak force generated by each subsequent tetanus. Simultaneously, we observed an increase in the baseline force (i.e. sustained force) between subsequent tetani (figure 2). Over the first 80 s, the force output during the individual tetanic contractions decreased precipitously (figure 2a). However, after approximately 60 s the muscle complex also failed to achieve complete relaxation between tetanic contractions, as evidenced by an increase in the sustained force between contractions (figure 2b). From approximately 140–300 s into the fatigue test, a sustained force clearly was present, with decreasing peak tetanic forces superimposed upon the sustained force (figure 2c). By 300 s into the test, sustained force nearly matches peak tetanic force, and the sustained force is maintained between tetanic trains when no stimulation is occurring (figure 2d,e). Males and females did not differ in maximum sustained force (observed at approx. 200 ms) relative to maximum peak tetanic force at the beginning of the test ($F = 1.4051$, d.f. = 1.7, $p = 0.2807$). For males, the mean ratio of maximum sustained force to maximum peak tetanic force was $0.248 \pm 0.032$ with a range of 0.209–0.344. For females, the mean ratio was $0.308 \pm 0.027$ with a range of 0.276–0.390. Thus, overall, maximum sustained bite force achieved during the *in situ* fatigue tests ranged from 20.9% to 39.0% of maximum tetanic bite force.

## (c) Myosin heavy chain isoform identification

On the electrophoretic gels, we observed three distinct bands for the thigh muscle sample and a single, large band that seems to be two diffuse bands for the jaw muscle sample (figure 3a,b). The top band for the jaw muscle separates between the second band (type II) and last band (type I) in the lizard thigh muscle thus indicating a separate myosin isoform (figure 3a,b). The second band below appears to line up with the type I slow fibre in the lizard thigh muscle, as well as the type I fibre in the mouse gastrocnemius. Western blot analysis indicates that there are type IIm (masticatory myosin) and type I myosin in the jaw muscle that reacted with MAb 2F4c (type IIm) and MAb ALD-58 (type I)

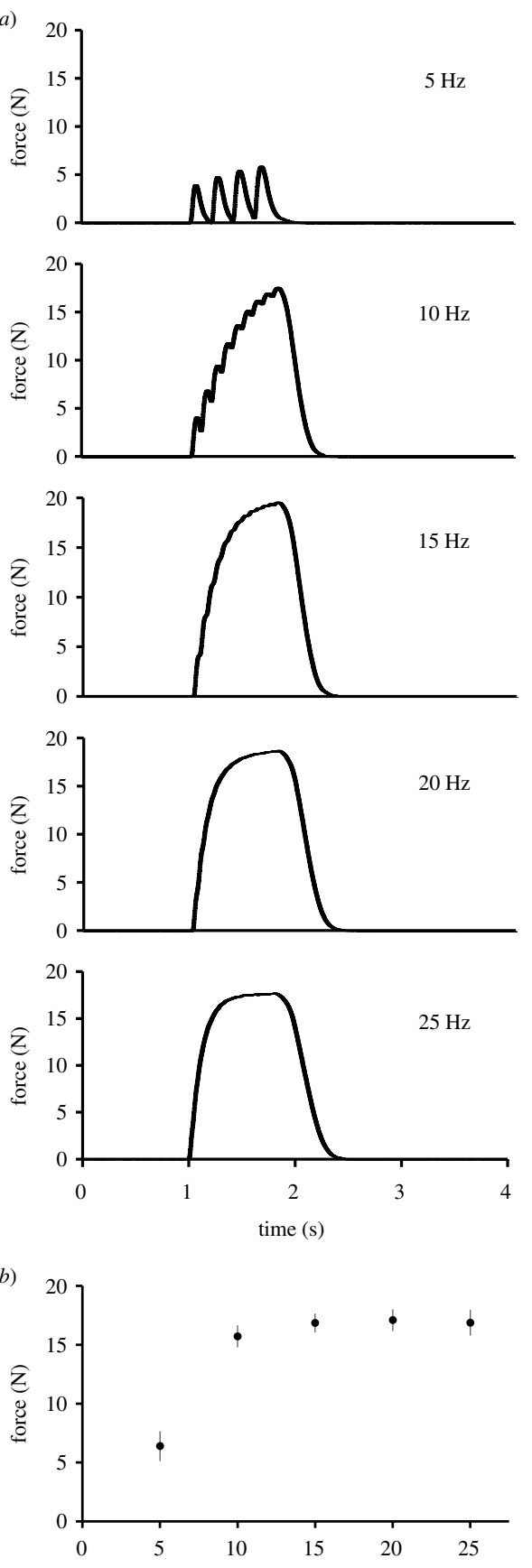

**Figure 1.** Force–frequency relationship. (*a*) A representative example of the force–frequency relationship of the jaw-adductor muscle complex of *E. multicarinata*. The muscle complex was stimulated bilaterally with supramaximal voltage for a 750 ms train duration with a pulse duration of 0.2 ms at frequencies ranging from 5 to 25 Hz. (*b*) Force–frequency curve of the jaw-adductor muscle complex of *E. multicarinata*. Values are means ± s.e.m. for all individuals combined at each pulse frequency (males: $n = 4$, females: $n = 4$).

*Proc. R. Soc. B* **287**: 20201578

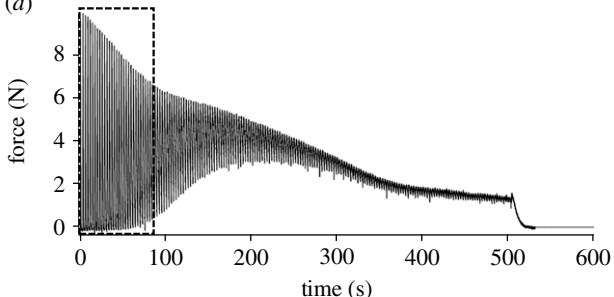

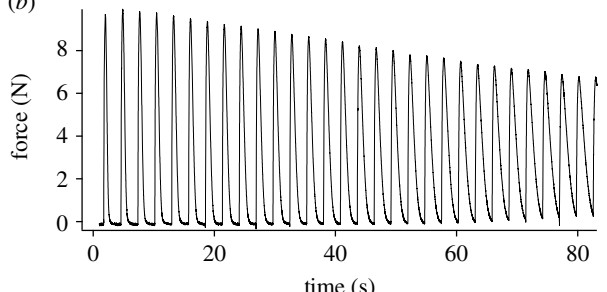

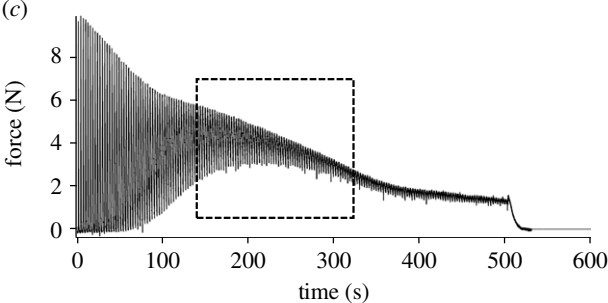

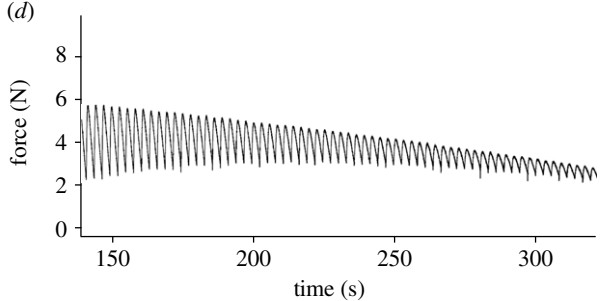

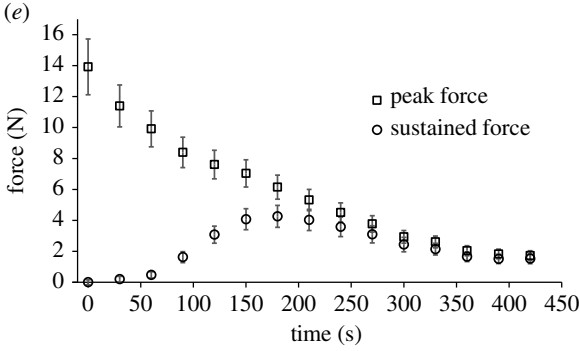

**Figure 2.** Fatigue test. Representative force profile of *E. multicarinata* jaw-adductor muscle complex directly stimulated at supramaximal voltage with a pulse duration of 0.2 ms at 60 Hz for a 150 ms tetanus repeated every 3 s. (*a*) Complete relaxation of individual tetanic contractions occurs for approximately first 60 s of fatigue test. (*b*) Expansion of approximately first 80 s of force profile of fatigue test showing individual tetanic contractions. Full relaxation of the muscle complex occurs until approximately 60 s into the test, after which sustained force develops. (*c*) Sustained force is at its maximum from approximately 150 to 250 ms. (*d*) Expansion of force profile showing maximum sustained force with superimposed peak tetanic forces. (*e*) Mean ± s.e.m. of peak tetanic force (squares) and sustained force (circles) from raw fatigue test (*a*) (males: $n = 4$, females: $n = 4$).

(figure 3*c,d*). The single-fibre SDS–PAGE gels showed two distinct bands in the jaw muscle (figure 3*e,f,g*).

## (d) Immunohistochemistry

Ten monoclonal antibodies (MAbs) allowed us to characterize six different patterns of MHC isoforms in the *E. multicarinata* thigh sample (electronic supplementary material, table S1). We recognized two different patterns of slow isoform and two different patterns of fast isoform for the 12 fibres that we identified from the serial cross sections (electronic supplementary material, table S1 and figure 4). The same MAbs were used for comparison with the jaw muscle samples. All the jaw muscle fibres tested reacted with three of the ten MAbs: 2F4c, A4.15.19 and ALD-58 (electronic supplementary material, table S2; figure 5).

## 4. Discussion

The objective of the current study was to characterize the contractile properties of the jaw-adductor muscle complex (i.e. adductor mandibulae complex) of the southern alligator lizard (*E. multicarinata*), with a primary focus on simulating *in situ* the natural *in vivo* function of the muscles (i.e. fatigue test). The results of our experiments show that the jaw-adductor muscle complex exhibits the contractile characteristics of a mix of muscle fibre types, as well as a fatigue-resistant property in which a sustained force is maintained for several minutes. Neither our experimental results, nor the unusual courtship behaviour, can be explained by known contractile properties typical of vertebrate skeletal muscle. Although the behaviour and the results of the *in situ* experiments both clearly indicate that the jaw-adductor complex exhibits specialized sustained force characteristics, the underlying morphological/physiological mechanism(s) is unknown. The behaviour suggests the presence of specialized muscle characteristics in the jaw-adductor muscles of *E. multicarinata*, such as a tonic isoform of MHC.

The experimentally characterized contractile properties indicate a mixed fibre type jaw-adductor muscle complex in *E. multicarinata*. The results of the force–frequency experiments point to characteristics of slow muscle fibres with complete tetanic fusion occurring at low frequencies of stimulation (20 Hz; figure 1*a*). Complete fusion of the jaw-adductor complex occurred at approximately half the stimulation frequency of that required for complete fusion in other 'slow' muscles. For example, the slow/tonic iliofibularis muscle of *Xenopus laevis* exhibited complete tetanic fusion at 40–60 Hz [38]. Relaxation times also indicate the presence of slow fibres in the jaw-adductor complex of *E. multicarinata*. Twitch ½RT of the jaw-adductor complex is relatively slow ($266 \pm 47.8$ ms) compared to, for example, the iliofibularis muscle in a generalized sprinting lizard (*Agama agama*, $81 \pm 15$ ms) or even a slow-moving arboreal chameleon (*Chamaeleo senegalensis*, $163 \pm 24$ ms), with the latter shown to have a large proportion tonic fibres in the iliofibularis muscle [28]. However, the jaw muscles of *E. multicarinata* are not as slow as the amplectic muscles of the extensor carpi radialis of a frog (*Rana temporaria*, $658 \pm 118$ ms) [15]. Conversely, the jaw-adductor muscle complex of *E. multicarinata* also exhibits characteristics of fast fibres (FG and/or FOG). The twitch TPT ($63.1 \pm 3.7$ ms) is moderately fast compared to, for example, the forelimb muscles and ankle extensor muscles

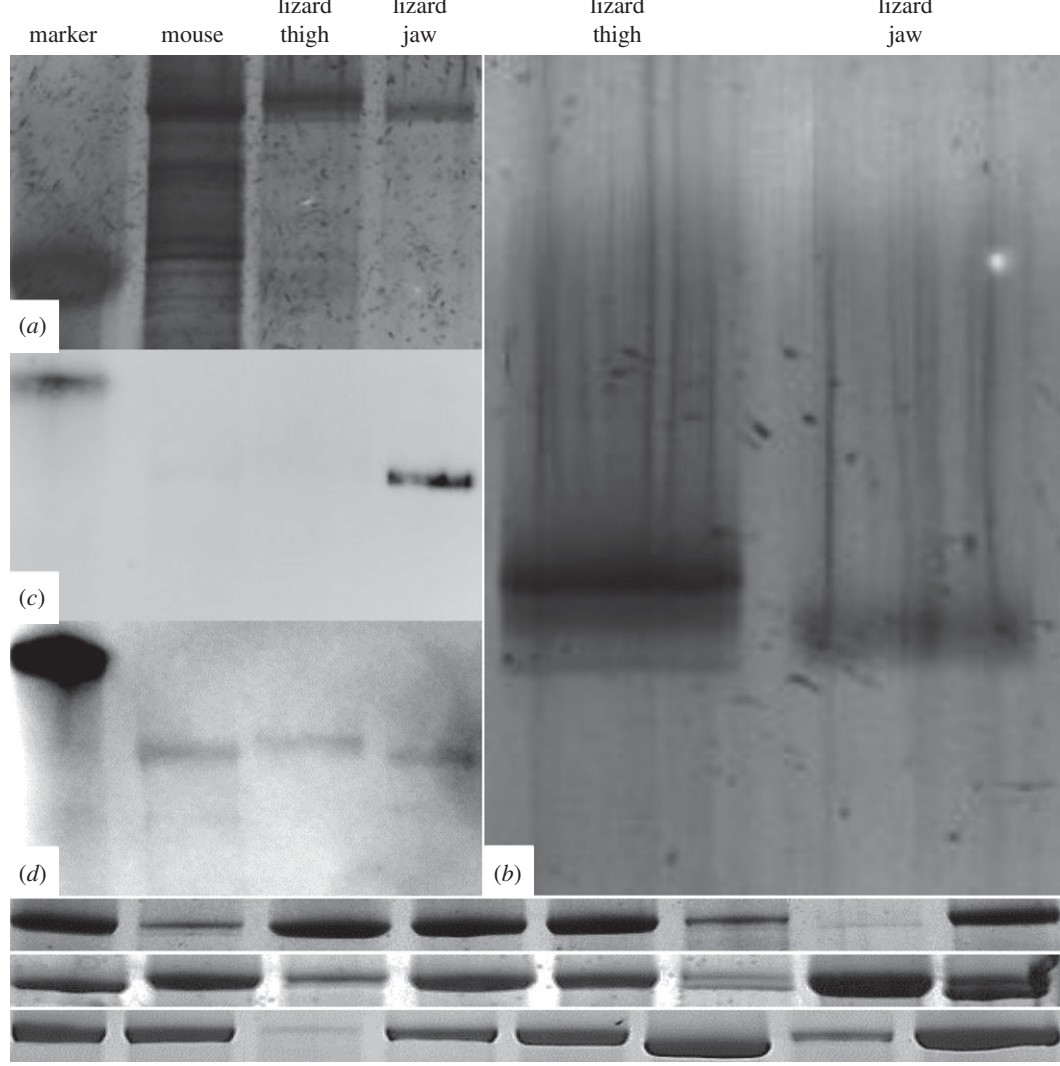

**Figure 3.** (*a*) Electrophoretic separation of MHC isoforms in mouse gastrocnemius, lizard thigh muscle and lizard jaw-adductor complex. (*b*) Zoomed in image of figure 3*a*. Electrophoretic separation of lizard thigh and jaw-adductor muscle samples. The jaw muscle seems to show two diffuse bands that represent a single, large band. (*c*) Western blot with MAb 2F4c (i.e. masticatory myosin). (*d*) Western blot with MAb ALD-58 (i.e. slow/tonic myosin). (*e–g*) Electrophoretic separation of single fibres from lizard adductor mandibular externus superficialis muscle. Each column shows the banding pattern of a separate individual fibre extracted from the muscle. Each row represents a different gel. Some of the columns from each gel show two distinct bands from the single-fibre electrophoresis thus indicating the presence of two MHC isoforms within individual fibres.

of *Rana (Lithobates) catesbeiana* that range from 58.2 ± 2.9 ms to 99.8 ± 3.1 ms [13,39] and the iliofibularis muscle of *A. agama* (58 ± 11 ms) [28]. In terms of fatiguability, the jaw-adductor muscle complex exhibited a steep decrease in the peak force produced by subsequent tetani over the first 80 s (figure 2*a*), which reflects the characteristics of fatiguing fast muscle fibres. This was followed by a more gradual decline in tetanic force output, likely due to fatiguing of slower muscle fibres, together with a rise in the sustained force developed in the muscle over time, as full relaxation is not achieved between the individual tetanic contractions.

The pattern of decreasing peak tetanic forces with a simultaneous increase in sustained inter-tetanic force suggests the presence of more than one fibre type, or different isoforms within a single-fibre type (figures 2 and 3), as well as the likely presence of a specialized mechanism to produce the sustained force capability. Among vertebrates, the ability to sustain a constant force has been observed in the forelimb muscles of males of some sexually dimorphic frogs, which are used by males to amplect females for extended periods. These muscles exhibit a similar sustained force during

*in situ* experiments [13–15]. In the frog system, this sustained force characteristic is only exhibited by the males, whereas for *E. multicarinata*, both sexes show this sustained force characteristic in the fatigue test. It is unknown whether the ability to generate sustained muscle force is the result of a similar mechanism(s) between these independently evolved systems.

Little has been published on the MHC isoforms of reptilian jaw muscles. Most work has been on mammalian systems and shows only masticatory myosin (IIm) in the jaw muscles. Hoh *et al.* [40] identified cat jaw muscle MHC isoforms through SDS–PAGE gels and western blots. The results based on single-fibre gel electrophoresis showed distinct 'jaw-fast' and 'jaw-slow' bands. Reiser & Bicer [41] tested various electrophoretic separation protocols on skeletal and cardiac muscle to identify MHC isoforms in reptilian muscles, including samples from four species of turtles (*Chelydra serpentina*, *Apalone spinifera*, *Chrysemys picta*, *Graptemys geographica*) and two species of lizards (*Varanus salvator*, *Corucia zebrata*). Notably, they found that the separation of the bands representing MHC isoforms distinctly differed between the jaw adductors and all other muscles tested in these reptiles.

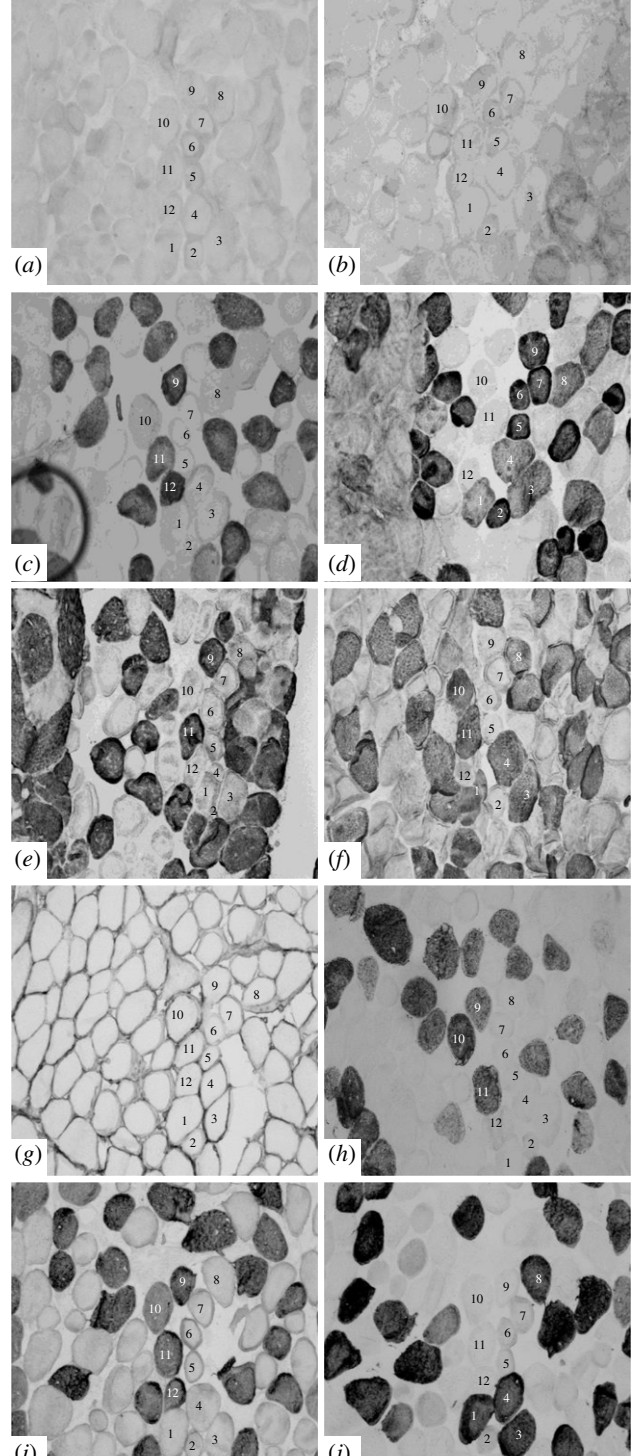

**Figure 4.** Serial cross sections of *E. multicarinata* thigh muscle stained with antibodies: (*a*) 2F4c (anti-MHC IIm), (*b*) SC-71 (anti-MHC IIa), (*c*) A4.15.19 (anti-MHC IIa), (*d*) ALD-58 (anti-MHC I), (*e*) BF-13 (anti-MHC IIa, IIx, IIb), (*f*) D5 (anti-MHC I), (*g*) BF-F3 (anti-MHC IIb), (*h*) N3.36 (neonatal & II), (*i*) Sigma Fast (anti-MHC IIa, IIx, IIb), (*j*) Vector Slow (anti-MHC I). Fibres 1, 3, 4, 8 exhibit pattern A slow characteristics. Fibres 2, 5, 6, 7 exhibit pattern B slow characteristics. Fibres 9, 10, 11, 12 exhibit four different patterns of fast characteristics.

Our electrophoretic, western blot and immunohistochemistry results suggest there is only one muscle fibre type with two MHC isoforms within the adductor mandibulae externus superficialis jaw muscle of *E. multicarinata* (figures 3–5). The single-fibre gels show two separate banding patterns for the jaw muscle (figure 3*e,f,g*). Western blot analysis shows

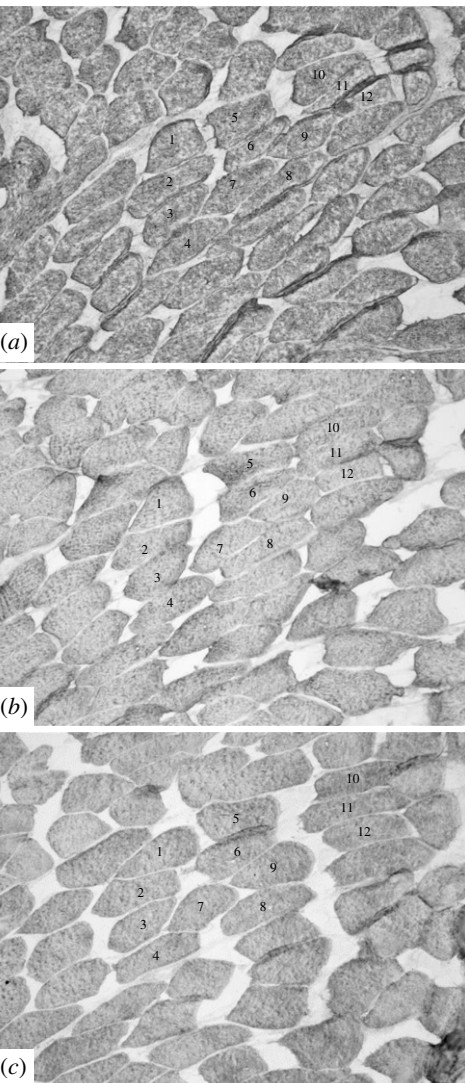

**Figure 5.** Serial cross sections of *E. multicarinata* jaw muscle stained with Mabs: (*a*) 2F4c (anti-MHC IIm), (*b*) A4.15.19 (anti-MHC IIa), (*c*) ALD-58 (anti-MHC I).

positive results for MAb 2F4c for the masticatory myosin isoform and ALD-58 for slow/tonic myosin isoforms, which suggests a more homogeneous fibre type within the jaw muscle in comparison to the thigh muscle based on the fibres we tested (figure 3*c,d*). The MAb 2F4c is known to react with masticatory myosin while ALD-58 has been used in chicken and turtle experiments to identify tonic muscle fibres [23]. The presence of the masticatory myosin isoform was not surprising, as it is found widely in jaw muscles in mammalian systems [40]. What was unexpected was the reaction with ALD-58 (figure 3*d*), which led us to investigate the jaw muscle further with single-fibre extractions (figure 3*e,f,g*) and immunohistochemistry (figures 4 and 5). We distinguished two different patterns of slow isoform and two different patterns of fast isoform for the 12 fibres that we identified from the serial cross sections for the lizard thigh muscle (electronic supplementary material, table S1 and figure 4). The same MAbs were used for comparison with the jaw muscle samples. The jaw muscle samples reacted with only three of the ten MAbs: 2F4c, A4.15.19 and ALD-58 (electronic supplementary material, table S2; figure 5). Further studies are needed to determine if these patterns are consistent with those for other vertebrates. The two

myosin isoforms found in the jaw muscle may contribute to the mixed-fibre type-like characteristics found in the contractile property experiments. The masticatory myosin may account for the fast fibre contractile characteristics, whereas the tonic isoform contributes to sustained inter-tetanic force. It is also possible that additional MHC isoforms are present that are indiscernible with the techniques used in this study, that another region of the examined jaw muscle includes other fibre types, or that other jaw-adductor muscles include other MHC isoforms. Future work will be necessary to identify the MHC isoforms in all the jaw muscles present within the complex, as well as outside the complex (i.e. pterygoideus) to determine whether they contribute to the sustained force phenomenon.

The results of our investigation into the contractile properties of the jaw muscles of *E. multicarinata* support our hypothesis that the jaw-adductor complex is specialized for generating significant sustained bite force even after peak tetanically generated bite force has greatly declined. The sustaining of force between stimulus trains by the jaw-adductor muscle complex of *E. multicarinata* may be explained by cross-bridges continuously forming but not disengaging due to available calcium within the myoplasm, as a result of less sarcoplasmic reticulum (SR) associated with tonic fibres. In a study by Franzini-Armstrong *et al.* [42], the anatomy of the SR and transverse (T) tubule configuration, which directly influences calcium release, was found to differ among diverse species (e.g. fish, frog, bird, dog). They found that the sizes of the calcium release units of the tonic muscle fibres of the toadfish swimbladder and the cruralis muscle of the frog were identical. The dyads and triads of the slow-tonic muscle fibres were composed of junctional domains of SR and T-tubule and sarcolemma that contained short, but multiple rows of feet. Some of these junctions were large, and the feet were not identical, but they found that it was a possible calcium trapping mechanism [42]. This could result in calcium being readily available in the cytosol allowing for cross-bridges to reengage as fast as they disengage. The reuptake of calcium from the myoplasm is necessary for muscle relaxation to occur. The calcium is transported back into the SR and happens with the aid of proteins embedded within its membrane (e.g. SERCA pumps), as well as calcium modulators (e.g. parvalbumin) found within the myoplasm. The number of these calcium regulatory proteins varies within the same fibre type, affecting the rate at which calcium is transported [43,44].

Male *E. multicarinata* may be using the slowing of relaxation with increased fatigue to their advantage, as the mechanism of muscle relaxation may provide a means to mitigate fatigue during high-endurance behaviours. Relaxation of a muscle involves cessation of calcium release from the SR and calcium reuptake by proteins (e.g. SERCA pumps, parvalbumin) associated with the SR [44–46]. Slowing of relaxation generally follows fatigue in skeletal muscle, which could be beneficial during a behaviour requiring sustained muscle force generation, as it results in a greater degree of tetanic fusion at lower stimulation frequencies. In a comparison of the contractile properties of leg and jaw muscles in *Anolis* lizards [44], jaw muscles were shown to dissipate tension more slowly than leg muscles, and thus the jaw muscles may rely more on sustained tension during *in vivo* behaviours. What is generally viewed as a negative effect of fatigue might be coopted as a beneficial function during courtship behaviour by *E. multicarinata*. We speculate that the most likely scenario is that relaxation slows due to muscle fatigue, and these slowing contractile dynamics are coupled with modifications to the calcium handling mechanism. The slow removal of calcium may allow for incomplete relaxation and the maintenance of sustained force.

Further studies are needed to explore the basic anatomy and structures associated with tonic fibres (e.g. SR) and how biochemical properties such as ATP usage, lactate production, glycogen concentration and calcium-cycling mechanisms may play a role in fatigue-resistant muscle function. Identifying additional examples of high resistance to muscle fatigue, particularly exceptional cases that often seem to be associated with specialized behaviours, could prove fruitful in characterizing possible mechanisms. By comparing fatigue resistance in different systems, as well as the kinds of extreme high-endurance behaviours that employ them, we may gain insight into whether independently evolved high-endurance behaviours demonstrate commonalities in their underlying mechanisms.

**Ethics.** All procedures were approved by the Institutional Animal Care and Use Committees at the University of California, Irvine (protocol #2013-3110-1), and at California State Polytechnic University, Pomona (protocol #17.014).

**Data accessibility.** Data are available from the Dryad Digital Repository: https://dx.doi.org/10.5061/dryad.d7wm37pzw [47].

**Authors' contributions.** A.N. and A.K.L. conceived the idea. J.P.B., E.A., A.K.L. and R.J.T. designed the experiments. A.N. carried out the experiments and drafted the manuscript. A.K.L. did the statistical analysis. A.K.L., E.A., R.J.T. and J.P.B. critically revised the manuscript. All authors gave final approval for publication and agree to be held accountable for the work performed therein.

**Competing interests.** We declare we have no competing interests.

**Funding.** This work was supported by a NSF BioTiER scholarship to A.N.

**Acknowledgements.** We thank J. Alexander and the support staff of the Cal Poly Pomona Vivarium for assistance with animal care, M. Allahdadi for assistance with tissue sectioning and immunohistochemistry, T. Marino for assistance collecting specimens, and N. Holt and J. Jellyman for their valuable comments on drafts of the manuscript.

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
