## [Reviewer comments · Proceedings of the Royal Society B: Biological Sciences]

Review History

RSPB-2020-1578.R0 (Original submission)

Review form: Reviewer 1

Recommendation

Accept with minor revision (please list in comments)

Scientific importance: Is the manuscript an original and important contribution to its field?

Excellent

General interest: Is the paper of sufficient general interest?

Excellent

Quality of the paper: Is the overall quality of the paper suitable?

Excellent

Is the length of the paper justified?

Yes

Should the paper be seen by a specialist statistical reviewer?

No

Do you have any concerns about statistical analyses in this paper? If so, please specify them explicitly in your report.

No

It is a condition of publication that authors make their supporting data, code and materials available - either as supplementary material or hosted in an external repository. Please rate, if applicable, the supporting data on the following criteria.

Is it accessible?

Yes

Is it clear?

Yes

Is it adequate?

Yes

Do you have any ethical concerns with this paper?

No

Comments to the Author

This article by Nguyen et al. presents an interesting study examining the fatigue resistance and myosin isoforms of jaw muscles in an anguid lizard. The authors present a very nicely performed and presented examination of the mechanisms underlying prolonged courtship biting behavior whereby they performed in situ bite force measurements while cyclically stimulating the jaw adductor muscles on both sides of the head. They then examined muscle fiber composition of these muscles. The results suggest that both tonic and masticatory myosin fibers are present in these jaw muscles and that characteristics of the bite force traces are likely related to the presence of each in the jaws. In particular, the declining peak tetanic forces appear to represent fast fiber contractile characteristics, possible from masticatory myosin, while the simultaneous increase in sustained inter-tetanic force are likely explained by the activity of tonic isoforms.

Overall I found the article to be very well written, the research well executed, and the results quite compelling. This article should provide valuable insight into specializations to muscle function based on their biological use, including compelling information about reduced fatigability in a vertebrate muscle, known previously only in a male frog amplexus muscles. While I note a couple very minor edits below, I find the manuscript as a whole to be excellent.

Specific comments:

Page 7-8, Lines 144-149 & 174-176: These sections initially appeared in my first reading to be inconsistent with regard to the origin of the single-fiber electrophoresis. This stemmed from the two sources listed in the first set of lines (two animals for which muscles first were used in immunohistochemistry and an animal for which muscles were only used in single-fiber electrophoresis), while the second set of lines only mentioned the first source. It might be worthwhile adding wording acknowledging both sources here too. Something to the effect of: "...the remaining jaw-adductor muscles from the immunohistochemistry experiments and the jaw-adductor muscles from a third individual were placed in..."

Page 8, Lines 185-190: Excellent methodology for testing for sex effects!

Page 11, Lines 263-270: *Agama stellio* is now classified as *Stellagama stellio* (Baig, Khalid Javed; Philipp Wagner, Natalia B. Ananjeva & Wolfgang Böhme 2012. A morphology-based taxonomic

revision of *Laudakia* Gray, 1845 (Squamata: Agamidae). *Vertebrate Zoology* 62 (2): 213-260.) and the authors should either include the updated genus in parentheses (as you have with *Rana* (*Lithobates*) *catesbeiana*) or the updated taxonomy adopted. That said, Abu-Ghalyun et al. (1988) indicate that *Agama agama* (rather than *stellio*) was used for their experiments. Further, the authors are missing an "a" in their spelling of "Chamaeleo".

Page 12, Line 304: Spelling should be "serpentina" ending with an "-a".

Review form: Reviewer 2

Recommendation

Accept with minor revision (please list in comments)

Scientific importance: Is the manuscript an original and important contribution to its field?

Good

General interest: Is the paper of sufficient general interest?

Excellent

Quality of the paper: Is the overall quality of the paper suitable?

Good

Is the length of the paper justified?

Yes

Should the paper be seen by a specialist statistical reviewer?

No

Do you have any concerns about statistical analyses in this paper? If so, please specify them explicitly in your report.

Yes

It is a condition of publication that authors make their supporting data, code and materials available - either as supplementary material or hosted in an external repository. Please rate, if applicable, the supporting data on the following criteria.

Is it accessible?

Yes

Is it clear?

Yes

Is it adequate?

Yes

Do you have any ethical concerns with this paper?

No

Comments to the Author

This is an interesting and novel paper that determines the isometric properties and fibre type of the adductor mandibulae muscle complex in the southern alligator lizard. The findings, that the muscle can maintain tonic type forces over a prolonged period of time and has a tonic fibre type, relate well to unpublished observations of prolonged biting behaviour by males, of females,

during courtship.

In the introduction there are a couple of comments made that need rewording as they make the findings illustrated seem more unique than they are – see minor comments below re lines 49-50 and 55-56. The following review of literature is more objective and undermines the validity of these statements.

As indicated below I think it would be useful to provide more details and citation of references in the muscle physiology methods section (in the same way they are cited in MHC isoform characterisation section) to help the reader to understand the methods, the underlying reasons for this methodological approach and to ensure that they are reproducible.

Minor comments

Lines 49-50: Are you able to cite references to support that there is an “established dogma on the physiological limitations, in terms of fatigue resistance, of vertebrate skeletal muscle”? As you later suggest there are other reports of behaviours in the literature such as amplexus in amphibians that are similar and a few of studies of the physiology of isolated tonic fibre bundles or muscles containing tonic fibres (particularly in fish and amphibians) that counter the wording used in the manuscript that suggests this is a new finding “constitutes a behavior so extreme that it appears to counter established dogma on the physiological limitations, in terms of fatigue resistance, of vertebrate skeletal muscle”. E.g. Tonic fibres have previously been identified in earlier histochemical studies on limb muscle in lizards e.g. Bonine et al 2005 and have been implicated in the clinging performance observed in chameleons.

Lines 55-56: “the ability of muscle to sustain constant force for long periods has been observed only in the amplexic muscles of males” – as mentioned above I believe that this is incorrect, as currently worded, as other studies have looked at fish tonic muscle fibres from jaw and elsewhere on the body and tonic muscle from other species’ jaw muscles.

Line 100: It might be useful to indicate how lizards were caught.

Line 101: so how long were lizards kept in captivity prior to testing?

Line 111: I suggest that you define any abbreviations on first use, such as ‘DI’, unless they are chemical symbols.

It would be useful in various parts of the methods to have further citation of literature to justify the approaches taken e.g. Ringer recipe used.

From my understanding of the methods there is no attempt to optimise the muscle length to maximise force production. Each lizards jaw is clamped around the force transducer so is a set gape distance but this could result in the sarcomere lengths of the jaw muscle tested in each lizard potentially being very different. It would be useful to know the body mass and body length of the lizards used to get an idea of whether they are similarly sized.

Line 113 and thereabouts: So if the Ringer solution was used to irrigate, suggesting it continues to flow over the muscle, then on to waste, and you maintained muscle temperature - was the room kept at constant temperature or the Ringer? Was the head held in place, if so how? It might be useful to have a figure of the experimental setup (possibly as a supplementary figure) or further description – e.g. how are the stimulator wires held in place? So how many Volts and Amps were used in each stimulation and how was this determined?

Line 128: you have indicated there was a period of rest between each tetanus but were any stimulation frequencies repeated to check for change in performance over time or to see if the rest period was sufficient? Was it a set order of stimulation frequency such that there might be an underlying order effect?

Lines 132-133: Why choose 25Hz for the stimulation frequency for tetanus times? – stimulation for peak tetanic force clearly varied between individuals. Where did the $\frac{1}{2}RT$ measurement commence from for the tetanus? – usually this would be last stimulus

Line 135: So which stimulation frequency was used for fatigue tests?

Line 195 and elsewhere in the results: you refer to Figure 1A, but Figure 1 does not have labelling of A, B or other sections, neither does the figure legend indicate that there are different subsections to this figure. Was there a difference in peak force between male and female? Could you convert the force to a stress (force/cross-sectional area) to account for potential differences in size between male and female?

Line 205 What do you mean by “tended”? Why not give a p value? Why conduct pairwise

comparisons if the ANOVA shows no difference?

Lines 209-223 It would be helpful if you cited the separate figures in figure 2 individually in this results text to lead the reader through your findings. You also need to label the subsections of the figure A, B etc on the figure.

Lines 239-240 Is it worth further clarifying that all jaw fibres tested reacted to all of these 3 antibodies suggesting a more homogenous fibre type in jaw than in thigh on the basis of the fibres you have tested, however recognising that there could be different fibre types untested in the muscle.

Line 264: last stimulus to half tetanus relaxation in *Rana temporaria* extensor carpi radialis muscle was reported as 658ms for males in Navas and James 2007. Not sure why you haven't cited this value but have cited the paper elsewhere in your manuscript.

Line 326: or that another region of the jaw muscle you used contains fibres of other fibre type.

Figure 3: have you got any better quality versions of these figures to be able to better discern the bands in the jaw muscle?

Figure 3 legend: under section B change reference to '4A' to '3A' I think we could do with some clearer explanation of what E, F and G represent.

Decision letter (RSPB-2020-1578.R0)

23-Jul-2020

Dear Dr Lappin:

Your manuscript has now been peer reviewed and the reviews have been assessed by an Associate Editor. The reviewers' comments (not including confidential comments to the Editor) and the comments from the Associate Editor are included at the end of this email for your reference. As you will see, the reviewers and the Editors have raised some concerns with your manuscript and we would like to invite you to revise your manuscript to address them.

Research ethics:

Use of animals and field studies:

It is a condition of publication that you make available the data and research materials supporting the results in the article. Please see our Data Sharing Policies (<https://royalsociety.org/journals/authors/author-guidelines/#data>). Datasets should be deposited in an appropriate publicly available repository and details of the associated accession number, link or DOI to the datasets must be included in the Data Accessibility section of the article (<https://royalsociety.org/journals/ethics-policies/data-sharing-mining/>). Reference(s) to datasets should also be included in the reference list of the article with DOIs (where available).

Please submit a copy of your revised paper within three weeks. If we do not hear from you within this time your manuscript will be rejected. If you are unable to meet this deadline please let us know as soon as possible, as we may be able to grant a short extension.

Best wishes,
Dr John Hutchinson, Editor
mailto: proceedingsb@royalsociety.org

Associate Editor

Comments to Author:

Thank you for the opportunity to review this manuscript. Reviews from two referees have now been received. Both referees viewed the MS very favorably, finding the study to be well-executed, and its findings of novel muscle fatigue resistance likely of significant importance and general interest. Both referees also recommended points to be considered for revision. In particular, Referee 2 expressed concern that some of the framing of the Introduction may have gone further than was warranted based on the Results; in addition, a number of additional points of clarification were requested for the Methods.

Considering these recommendations, I encourage you to submit a revised version of the manuscript that addresses the comments provided by the referees. In addition, I have listed some additional minor corrections that should be addressed. I would further recommend that, in conclusions to the Abstract and Discussion in particular, it would be good broaden out somewhat beyond the focus on the peculiar incidence of fatigue resistance traits in this species, and to reach out a bit to articulate what their identification means for understanding muscle function more generally. This is a distinct issue from that raised by Referee 2, who expressed concern that specific statements may have been too broad. The latter point I have just noted recommends making additional effort to place the results of this study in a broader context.

Thank you once again for your submission.

Additional recommended minor corrections:

L64. Change “slow” to “slower”.

L65. Remove “they”.

L295. Year for Navas and James citation is 2007 in the text, but listed as 2006 in the literature cited.

L325. Change “contribute” to “contributes”.

L351. Remove “the” before “muscle”.

Reviewer(s)' Comments to Author:

Referee: 1

Comments to the Author(s)

This article by Nguyen et al. presents an interesting study examining the fatigue resistance and myosin isoforms of jaw muscles in an anguid lizard. The authors present a very nicely performed and presented examination of the mechanisms underlying prolonged courtship biting behavior whereby they performed in situ bite force measurements while cyclically stimulating the jaw adductor muscles on both sides of the head. They then examined muscle fiber composition of these muscles. The results suggest that both tonic and masticatory myosin fibers are present in these jaw muscles and that characteristics of the bite force traces are likely related to the presence of each in the jaws. In particular, the declining peak tetanic forces appear to represent fast fiber contractile characteristics, possible from masticatory myosin, while the simultaneous increase in sustained inter-tetanic force are likely explained by the activity of tonic isoforms.

Overall I found the article to be very well written, the research well executed, and the results quite compelling. This article should provide valuable insight into specializations to muscle function based on their biological use, including compelling information about reduced

fatigability in a vertebrate muscle, known previously only in a male frog amplexus muscles. While I note a couple very minor edits below, I find the manuscript as a whole to be excellent.

Specific comments:

Page 7-8, Lines 144-149 & 174-176: These sections initially appeared in my first reading to be inconsistent with regard to the origin of the single-fiber electrophoresis. This stemmed from the two sources listed in the first set of lines (two animals for which muscles first were used in immunohistochemistry and an animal for which muscles were only used in single-fiber electrophoresis), while the second set of lines only mentioned the first source. It might be worthwhile adding wording acknowledging both sources here too. Something to the effect of: "...the remaining jaw-adductor muscles from the immunohistochemistry experiments and the jaw-adductor muscles from a third individual were placed in..."

Page 8, Lines 185-190: Excellent methodology for testing for sex effects!

Page 11, Lines 263-270: *Agama stellio* is now classified as *Stellagama stellio* (Baig, Khalid Javed; Philipp Wagner, Natalia B. Ananjeva & Wolfgang Böhme 2012. A morphology-based taxonomic revision of *Laudakia* Gray, 1845 (Squamata: Agamidae). *Vertebrate Zoology* 62 (2): 213-260.) and the authors should either include the updated genus in parentheses (as you have with *Rana (Lithobates) catesbeiana*) or the updated taxonomy adopted. That said, Abu-Ghalyun et al. (1988) indicate that *Agama agama* (rather than *stellio*) was used for their experiments. Further, the authors are missing an "a" in their spelling of "Chamaeleo".

Page 12, Line 304: Spelling should be "serpentina" ending with an "-a".

Referee: 2

Comments to the Author(s)

This is an interesting and novel paper that determines the isometric properties and fibre type of the adductor mandibulae muscle complex in the southern alligator lizard. The findings, that the muscle can maintain tonic type forces over a prolonged period of time and has a tonic fibre type, relate well to unpublished observations of prolonged biting behaviour by males, of females, during courtship.

In the introduction there are a couple of comments made that need rewording as they make the findings illustrated seem more unique than they are – see minor comments below re lines 49-50 and 55-56. The following review of literature is more objective and undermines the validity of these statements.

As indicated below I think it would be useful to provide more details and citation of references in the muscle physiology methods section (in the same way they are cited in MHC isoform characterisation section) to help the reader to understand the methods, the underlying reasons for this methodological approach and to ensure that they are reproducible.

Minor comments

Lines 49-50: Are you able to cite references to support that there is an "established dogma on the physiological limitations, in terms of fatigue resistance, of vertebrate skeletal muscle"? As you later suggest there are other reports of behaviours in the literature such as amplexus in amphibians that are similar and a few of studies of the physiology of isolated tonic fibre bundles or muscles containing tonic fibres (particularly in fish and amphibians) that counter the wording used in the manuscript that suggests this is a new finding "constitutes a behavior so extreme that it appears to counter established dogma on the physiological limitations, in terms of fatigue resistance, of vertebrate skeletal muscle". E.g. Tonic fibres have previously been identified in earlier histochemical studies on limb muscle in lizards e.g. Bonine et al 2005 and have been implicated in the clinging performance observed in chameleons.

Lines 55-56: "the ability of muscle to sustain constant force for long periods has been observed only in the amplexic muscles of males" – as mentioned above I believe that this is incorrect, as

currently worded, as other studies have looked at fish tonic muscle fibres from jaw and elsewhere on the body and tonic muscle from other species' jaw muscles.

Line 100: It might be useful to indicate how lizards were caught.

Line 101: so how long were lizards kept in captivity prior to testing?

Line 111: I suggest that you define any abbreviations on first use, such as 'DI', unless they are chemical symbols.

It would be useful in various parts of the methods to have further citation of literature to justify the approaches taken e.g. Ringer recipe used.

From my understanding of the methods there is no attempt to optimise the muscle length to maximise force production. Each lizards jaw is clamped around the force transducer so is a set gape distance but this could result in the sarcomere lengths of the jaw muscle tested in each lizard potentially being very different. It would be useful to know the body mass and body length of the lizards used to get an idea of whether they are similarly sized.

Line 113 and thereabouts: So if the Ringer solution was used to irrigate, suggesting it continues to flow over the muscle, then on to waste, and you maintained muscle temperature - was the room kept at constant temperature or the Ringer? Was the head held in place, if so how? It might be useful to have a figure of the experimental setup (possibly as a supplementary figure) or further description - e.g. how are the stimulator wires held in place? So how many Volts and Amps were used in each stimulation and how was this determined?

Line 128: you have indicated there was a period of rest between each tetanus but were any stimulation frequencies repeated to check for change in performance over time or to see if the rest period was sufficient? Was it a set order of stimulation frequency such that there might be an underlying order effect?

Lines 132-133: Why choose 25Hz for the stimulation frequency for tetanus times? - stimulation for peak tetanic force clearly varied between individuals. Where did the $\frac{1}{2}RT$ measurement commence from for the tetanus? - usually this would be last stimulus

Line 135: So which stimulation frequency was used for fatigue tests?

Line 195 and elsewhere in the results: you refer to Figure 1A, but Figure 1 does not have labelling of A, B or other sections, neither does the figure legend indicate that there are different subsections to this figure. Was there a difference in peak force between male and female? Could you convert the force to a stress (force/cross-sectional area) to account for potential differences in size between male and female?

Line 205 What do you mean by "tended"? Why not give a p value? Why conduct pairwise comparisons if the ANOVA shows no difference?

Lines 209-223 It would be helpful if you cited the separate figures in figure 2 individually in this results text to lead the reader through your findings. You also need to label the subsections of the figure A, B etc on the figure.

Lines 239-240 Is it worth further clarifying that all jaw fibres tested reacted to all of these 3 antibodies suggesting a more homogenous fibre type in jaw than in thigh on the basis of the fibres you have tested, however recognising that there could be different fibre types untested in the muscle.

Line 264: last stimulus to half tetanus relaxation in *Rana temporaria* extensor carpi radialis muscle was reported as 658ms for males in Navas and James 2007. Not sure why you haven't cited this value but have cited the paper elsewhere in your manuscript.

Line 326: or that another region of the jaw muscle you used contains fibres of other fibre type.

Figure 3: have you got any better quality versions of these figures to be able to better discern the bands in the jaw muscle?

Figure 3 legend: under section B change reference to '4A' to '3A' I think we could do with some clearer explanation of what E, F and G represent.

Author's Response to Decision Letter for (RSPB-2020-1578.R0)

See Appendix A.

RSPB-2020-1578.R1 (Revision)

Review form: Reviewer 2

Recommendation

Accept with minor revision (please list in comments)

Scientific importance: Is the manuscript an original and important contribution to its field?

Good

General interest: Is the paper of sufficient general interest?

Good

Quality of the paper: Is the overall quality of the paper suitable?

Good

Is the length of the paper justified?

Yes

Should the paper be seen by a specialist statistical reviewer?

No

Do you have any concerns about statistical analyses in this paper? If so, please specify them explicitly in your report.

No

It is a condition of publication that authors make their supporting data, code and materials available - either as supplementary material or hosted in an external repository. Please rate, if applicable, the supporting data on the following criteria.

Is it accessible?

Yes

Is it clear?

Yes

Is it adequate?

Yes

Do you have any ethical concerns with this paper?

No

Comments to the Author

The authors have done a good job of responding to most of the comments that I have made and overall the manuscript has improved.

I have a few minor suggestions in reply to the authors' response to my original comments that I think, in particular, will further clarify the methods:

Line 160: Before the sentence beginning with "Once" I suggest you add a couple of further sentences to help clarify the methods further - "No attempt was made to identify the gape angle at which the jaw muscles would produce maximum force as we were focused primarily on the fatigue characteristics of the jaw-adductor complex, and not maximum force production. Also,

the relatively small variation in head size of the individuals used should have limited variation in the relative starting length of the muscle."

Line 164: I'd suggest altering "We allowed the muscles to rest for two minutes between each stimulus train to minimize fatigue." to "We allowed the muscles to rest for two minutes between each stimulus train to minimize fatigue, as preliminary experiments indicated that this rest duration was sufficient for the muscle complex to recover and repeatedly achieve a similar force output."

Line 168: I suggest that you add the following sentence "We used 25 Hz for the stimulation frequency because by this frequency tetanic fusion had occurred in all muscle preparations tested."

Line 171: I suggest that you add the following sentence "We used a stimulation frequency of 60 Hz for the fatigue tests to ensure that all of the fibers in the muscle complex were being fully activated."

Is Figure 2E cited anywhere?

Decision letter (RSPB-2020-1578.R1)

21-Aug-2020

Dear Dr Lappin

I am pleased to inform you that your Review manuscript RSPB-2020-1578.R1 entitled "Fatigue resistant jaw muscles facilitate long-lasting courtship behavior in the southern alligator lizard (*Elgaria multicarinata*)" has been accepted for publication in Proceedings B. Congratulations!!

The referee(s) do not recommend any further changes. Therefore, please proof-read your manuscript carefully and upload your final files for publication. Because the schedule for publication is very tight, it is a condition of publication that you submit the revised version of your manuscript within 7 days. If you do not think you will be able to meet this date please let me know immediately.

To upload your manuscript, log into <http://mc.manuscriptcentral.com/prsb> and enter your Author Centre, where you will find your manuscript title listed under "Manuscripts with Decisions." Under "Actions," click on "Create a Revision." Your manuscript number has been appended to denote a revision.

You will be unable to make your revisions on the originally submitted version of the manuscript. Instead, upload a new version through your Author Centre.

- 1) A text file of the manuscript (doc, txt, rtf or tex), including the references, tables (including captions) and figure captions. Please remove any tracked changes from the text before submission. PDF files are not an accepted format for the "Main Document".
- 2) A separate electronic file of each figure (tiff, EPS or print-quality PDF preferred). The format should be produced directly from original creation package, or original software format. Please note that PowerPoint files are not accepted.

3) Electronic supplementary material: this should be contained in a separate file from the main text and the file name should contain the author's name and journal name, e.g. `authorname_procb_ESM_figures.pdf`

All supplementary materials accompanying an accepted article will be treated as in their final form. They will be published alongside the paper on the journal website and posted on the online figshare repository. Files on figshare will be made available approximately one week before the accompanying article so that the supplementary material can be attributed a unique DOI. Please see: <https://royalsociety.org/journals/authors/author-guidelines/>

4) Data-Sharing and data citation

It is a condition of publication that data supporting your paper are made available. Data should be made available either in the electronic supplementary material or through an appropriate repository. Details of how to access data should be included in your paper. Please see <https://royalsociety.org/journals/ethics-policies/data-sharing-mining/> for more details.

<http://datadryad.org/submit?journalID=RSPB&manu=RSPB-2020-1578.R1> which will take you to your unique entry in the Dryad repository.

Once again, thank you for submitting your manuscript to Proceedings B and I look forward to receiving your final version. If you have any questions at all, please do not hesitate to get in touch.

Sincerely,

Dr John Hutchinson

Associate Editor Board Member: 1

Comments to Author:

Thank you for submitting a revised version of your manuscript. We have received comments from one of the original referees, who indicated that the new version of the MS did a thorough job of addressing the comments raised in the original review. The referee noted a few remaining points that would benefit from clarifying revisions, the details of which are provided in the review.

Considering these recommendations, I encourage you to submit a revised version of the manuscript that addresses the comments provided by the referee. In addition, I list below two additional minor corrections that should be addressed. Thank you once again for your submission.

L206. Remove "at" before "5 Hz".

L342. Change "vary" to "varies" to correspond with "The number"; also add a comma after "fiber type".

Reviewer(s)' Comments to Author:

Referee: 2

Comments to the Author(s)

The authors have done a good job of responding to most of the comments that I have made and overall the manuscript has improved.

I have a few minor suggestions in reply to the authors' response to my original comments that I think, in particular, will further clarify the methods:

Line 160: Before the sentence beginning with "Once" I suggest you add a couple of further sentences to help clarify the methods further - "No attempt was made to identify the gape angle at which the jaw muscles would produce maximum force as we were focused primarily on the fatigue characteristics of the jaw-adductor complex, and not maximum force production. Also, the relatively small variation in head size of the individuals used should have limited variation in the relative starting length of the muscle."

Line 164: I'd suggest altering "We allowed the muscles to rest for two minutes between each stimulus train to minimize fatigue." to "We allowed the muscles to rest for two minutes between each stimulus train to minimize fatigue, as preliminary experiments indicated that this rest duration was sufficient for the muscle complex to recover and repeatedly achieve a similar force output."

Line 168: I suggest that you add the following sentence "We used 25 Hz for the stimulation frequency because by this frequency tetanic fusion had occurred in all muscle preparations tested."

Line 171: I suggest that you add the following sentence "We used a stimulation frequency of 60 Hz for the fatigue tests to ensure that all of the fibers in the muscle complex were being fully activated."

Is Figure 2E cited anywhere?

Decision letter (RSPB-2020-1578.R2)

01-Sep-2020

Dear Dr Lappin

I am pleased to inform you that your manuscript entitled "Fatigue resistant jaw muscles facilitate long-lasting courtship behavior in the southern alligator lizard (*Elgaria multicarinata*)" has been accepted for publication in Proceedings B.

Open Access

Paper charges

Sincerely,

Appendix A

Associate Editor Comments for the Author:

COMMENT (from Associate Editor summary): In particular, Referee 2 expressed concern that some of the framing of the Introduction may have gone further than was warranted based on the Results; in addition, a number of additional points of clarification were requested for the Methods.

RESPONSE: Please see our responses to the referee below. We have made changes to the wording, organization, and citations that we hope addresses the concerns.

COMMENT (from Associate Editor summary): I would further recommend that, in conclusions to the Abstract and Discussion in particular, it would be good broaden out somewhat beyond the focus on the peculiar incidence of fatigue resistance traits in this species, and to reach out a bit to articulate what their identification means for understanding muscle function more generally.

RESPONSE: Thank you for the suggestion. We have added text to the Abstract and Discussion in an attempt to broaden the language to point out potential implications of our results to muscle function in general instead of focusing solely on the jaw-adductor muscles of *E. multicaudata*. A point we make is that comparing the results of studies like ours to those from (past and future) studies of similar phenomena may help identify general physiological solutions to similar functional challenges.

Introduction

COMMENT: L64. Change “slow” to “slower”.

RESPONSE: Change made.

COMMENT: L65. Remove “they”.

RESPONSE: Change made.

Discussion

COMMENT: L295. Year for Navas and James citation is 2007 in the text, but listed as 2006 in the literature cited.

RESPONSE: Thank you for catching that! Change made in literature cited to correct the year to 2007. Also corrected in addition citation in text (Introduction).

COMMENT: L325. Change “contribute” to “contributes”.

RESPONSE: Change made.

COMMENT: L351. Remove “the” before “muscle”.

RESPONSE: Change made.

Referee 1 Comments for the Author:

This article by Nguyen et al. presents an interesting study examining the fatigue resistance and myosin isoforms of jaw muscles in an anguillid lizard. The authors present a very nicely performed and presented examination of the mechanisms underlying prolonged courtship biting behavior whereby they performed in situ bite force measurements while cyclically stimulating the jaw adductor muscles on both sides of the head. They then examined muscle fiber composition of these muscles. The results suggest that both tonic and masticatory myosin fibers are present in these jaw muscles and that characteristics of the bite force traces are likely related to the presence of each in the jaws. In particular, the declining peak tetanic forces appear to represent fast

fiber contractile characteristics, possible from masticatory myosin, while the simultaneous increase in sustained inter-tetanic force are likely explained by the activity of tonic isoforms.

Overall I found the article to be very well written, the research well executed, and the results quite compelling. This article should provide valuable insight into specializations to muscle function based on their biological use, including compelling information about reduced fatigability in a vertebrate muscle, known previously only in a male frog amplexus muscles. While I note a couple very minor edits below, I find the manuscript as a whole to be excellent.

Methods

COMMENT: Page 7-8, Lines 144-149 & 174-176: These sections initially appeared in my first reading to be inconsistent with regard to the origin of the single-fiber electrophoresis. This stemmed from the two sources listed in the first set of lines (two animals for which muscles first were used in immunohistochemistry and an animal for which muscles were only used in single-fiber electrophoresis), while the second set of lines only mentioned the first source. It might be worthwhile adding wording acknowledging both sources here too. Something to the effect of: "...the remaining jaw-adductor muscles from the immunohistochemistry experiments and the jaw-adductor muscles from a third individual were placed in..."

RESPONSE: Thank you for noting this discrepancy. We added text indicating that a third individual, not also used for the immunohistochemistry experiments, was included in the specimen sample for the single-fiber electrophoresis.

Discussion

COMMENT: Page 11, Lines 263-270: *Agama stellio* is now classified as *Stellagama stellio* (Baig, Khalid Javed; Philipp Wagner, Natalia B. Ananjeva & Wolfgang Böhme 2012. A morphology-based taxonomic revision of *Laudakia* Gray, 1845 (Squamata: Agamidae). *Vertebrate Zoology* 62 (2): 213-260.) and the authors should either include the updated genus in parentheses (as you have with *Rana (Lithobates) catesbeiana*) or the updated taxonomy adopted. That said, Abu-Ghalyun et al. (1988) indicate that *Agama agama* (rather than *stellio*) was used for their experiments. Further, the authors are missing an "a" in their spelling of "Chamaeleo".

RESPONSE: Thank you for catching these errors. Because Abu-Ghalyun et al. (1988) used *Agama agama* in their experiments, we changed the species from "*stellio*" to "*agama*" to accurately reflect this. We also corrected the spelling of *Chamaeleo*.

COMMENT: Page 12, Line 304: Spelling should be "serpentina" ending with an "-a".

RESPONSE: Corrected. Thank you.

Referee 2 Comments for the Author:

This is an interesting and novel paper that determines the isometric properties and fibre type of the adductor mandibulae muscle complex in the southern alligator lizard. The findings, that the muscle can maintain tonic type forces over a prolonged period of time and has a tonic fibre type, relate well to unpublished observations of prolonged biting behaviour by males, of females, during courtship.

In the introduction there are a couple of comments made that need rewording as they make the findings illustrated seem more unique than they are – see minor comments below re lines 49-50 and 55-56. The following review of literature is more objective and undermines the validity of these statements.

As indicated below I think it would be useful to provide more details and citation of references in the muscle physiology methods section (in the same way they are cited in MHC isoform characterisation section) to help

the reader to understand the methods, the underlying reasons for this methodological approach and to ensure that they are reproducible.

Introduction

COMMENT: Lines 49-50: Are you able to cite references to support that there is an “established dogma on the physiological limitations, in terms of fatigue resistance, of vertebrate skeletal muscle”? As you later suggest there are other reports of behaviours in the literature such as amplexus in amphibians that are similar and a few of studies of the physiology of isolated tonic fibre bundles or muscles containing tonic fibres (particularly in fish and amphibians) that counter the wording used in the manuscript that suggests this is a new finding “constitutes a behavior so extreme that it appears to counter established dogma on the physiological limitations, in terms of fatigue resistance, of vertebrate skeletal muscle”. E.g. Tonic fibres have previously been identified in earlier histochemical studies on limb muscle in lizards e.g. Bonine et al 2005 and have been implicated in the clinging performance observed in chameleons.

RESPONSE: We agree in retrospect that the wording of the original sentence the reviewer cites was too strong. We have reworded the sentence to soften the language (now first sentence of 2nd paragraph in introduction) and reorganized the introduction to more accurately place the mate-holding behavior in the context of previous studies. We initially focus on the studies of muscles involved with amplexus in frogs, as they show the most obvious parallels with our study. We also have added citations for studies of fatigue resistant capabilities in several vertebrates in the paragraph on the occurrence and functions tonic fibers (4th paragraph of introduction).

COMMENT: Lines 55-56: “the ability of muscle to sustain constant force for long periods has been observed only in the amplexic muscles of males” – as mentioned above I believe that this is incorrect, as currently worded, as other studies have looked at fish tonic muscle fibres from jaw and elsewhere on the body and tonic muscle from other species’ jaw muscles.

RESPONSE: As above, we agree and thank the reviewer for pointing this out. We have removed the sentence and added a couple of relevant references.

Methods

COMMENT: Line 100: It might be useful to indicate how lizards were caught.

RESPONSE: Change made. We have added that lizards were caught by hand.

COMMENT: Line 101: so how long were lizards kept in captivity prior to testing?

RESPONSE: Lizards were maintained in captivity for up to a year prior to being used in experiments. We have added this to the first paragraph of the methods. We also have added statements indicating that the lizards all were collected as adults, that they are easy to maintain in captivity, and that they were maintained in excellent condition until used in experiments.

COMMENT: Line 111: I suggest that you define any abbreviations on first use, such as ‘DI’, unless they are chemical symbols.

RESPONSE: “DI” changed to “distilled”.

COMMENT: It would be useful in various parts of the methods to have further citation of literature to justify the approaches taken e.g. Ringer recipe used.

RESPONSE: Thank you for the suggestion. We added a citation for the lizard Ringer’s solution.

COMMENT: From my understanding of the methods there is no attempt to optimise the muscle length to maximise force production. Each lizards jaw is clamped around the force transducer so is a set gape distance but this could result in the sarcomere lengths of the jaw muscle tested in each lizard potentially being very

different. It would be useful to know the body mass and body length of the lizards used to get an idea of whether they are similarly sized.

RESPONSE: This is a great comment. Because we were focused primarily on the fatigue characteristics of the jaw-adductor complex, and not maximum force production, we did not attempt to identify the gape angle at which the jaw muscles would produce maximum force. Although this would be quite interesting to look into, it also would be quite challenging for a number of reasons. First, the jaw-adductor complex is a triangular-shaped, compact, multi-pennate muscle group that would make it difficult to pinpoint optimum length of the fibers. Second, within the complex there are several muscles with some having additional subdivisions. The different muscles/subdivisions most likely achieve maximum force at lengths that correspond with different gape angles; a gape angle that optimizes the length of one muscle for force likely would put at least some other muscles away from their optimum length. In addition, if we consider the behavior, mate holding occurs at various gape angles, depending on the size of the male's head, the size of the female's head, the exact position of the hold, etc. Nevertheless, the specimens all were adults that did not vary greatly in size, and we made sure that the jaw placement on the transducer beams was consistent among experiments and constant within experiments. To address variation in size, we added summary statistics (mean \pm SEM) for snout-vent length, body mass, and head length, width, and depth in the text for the full sample, males only, and females only. The raw data will be included in the data repository.

COMMENT: Line 113 and thereabouts: So if the Ringer solution was used to irrigate, suggesting it continues to flow over the muscle, then on to waste, and you maintained muscle temperature - was the room kept at constant temperature or the Ringer?

RESPONSE: The room was kept at a constant temperature. We have added this information.

COMMENT: Was the head held in place, if so how? It might be useful to have a figure of the experimental setup (possibly as a supplementary figure) or further description – e.g. how are the stimulator wires held in place?

RESPONSE: We have added further description of positioning of the specimen (i.e., specimen positioned on a platform with adjacent transducer on clamp). We found that once the teeth engaged the leather strips on the ends of the transducer bars during the first stimulation, the jaws remained in a consistent position for the duration of the experiment. We also found that the electrodes consistently remained in place via the 3.5 mm long two two-pronged electrode tips embedded into the muscle complex on each side of the head.

COMMENT: So how many Volts and Amps were used in each stimulation and how was this determined?

RESPONSE: We found that the voltage that produced supramaximal stimulation ranged from 10-15 V. We were operating with a resistance of 250 ohms (a constant set on the box), so our current range was 0.04-0.06 amps (e.g., $I = V/R = 10/250 = 0.04$ amps). We have added this information to the text.

COMMENT: Line 128: you have indicated there was a period of rest between each tetanus but were any stimulation frequencies repeated to check for change in performance over time or to see if the rest period was sufficient? Was it a set order of stimulation frequency such that there might be an underlying order effect?

RESPONSE: In preliminary experiments, we found that a 2-minute rest between tetanic trains was sufficient for the muscle complex to apparently recover and repeatedly achieve a similar force output. However, we did not specifically test the effects of various rest periods once we saw similar forces after 2 minutes. The order of stimulation frequencies was consistently from lowest to highest, so we cannot completely dismiss a possible order effect. This said, our primary focus of the force-frequency experiments was to determine the tetanic fusion frequency of the muscle complex as a whole, and the results suggest that the complex is dominated by slow fibers. Moreover, the rapid decline in tetanic peak forces early in the fatigue test that followed the force-frequency experiment indicates that fast fibers were still being activated and subsequently fatigued.

COMMENT: Lines 132-133: Why choose 25Hz for the stimulation frequency for tetanus times? – stimulation for peak tetanic force clearly varied between individuals. Where did the $\frac{1}{2}$ RT measurement commence from for the tetanus? – usually this would be last stimulus.

RESPONSE: Tetanic time to peak tension (TPT) and half-relaxation time ($\frac{1}{2}$ RT) did vary some between individuals. We used 25 Hz for the stimulation frequency because at this frequency complete tetanus occurred. For the tetanus, the $\frac{1}{2}$ RT measurement commenced from the peak of tetanic tension to the point at which the force was half of the peak value. In the text, we state that the tetanic $\frac{1}{2}$ RT measurement was analogous to that for the twitch $\frac{1}{2}$ RT measurement.

COMMENT: Line 135: So which stimulation frequency was used for fatigue tests?

RESPONSE: We used a stimulation frequency of 60 Hz for the fatigue tests. We decided to increase the frequency well beyond what we observed was necessary for tetanic fusion (20-25 Hz) in an attempt to ensure that all of the fibers in the muscle complex were being fully activated.

Results

COMMENT: Line 195 and elsewhere in the results: you refer to Figure 1A, but Figure 1 does not have labelling of A, B or other sections, neither does the figure legend indicate that there are different subsections to this figure.

RESPONSE: Thank you for catching that! Corrections made to figure and figure legend.

COMMENT: Was there a difference in peak force between male and female?

RESPONSE: Mean peak tetanic force (25 Hz) was 18.2N for males and 15.6N for females. We ran an unpaired t-test and found no significant difference between the sexes. We would prefer not to include this information because to properly test for a sex difference in bite force we feel that a much larger sample would be desirable. A larger sample would take into account natural variation due to body size, musculo-skeletal size/shape, ontogeny, population differences, etc. We have found in other work in progress with this species that males have larger and more robust heads than females (absolutely and relative to body size) and bite significantly harder.

COMMENT: Could you convert the force to a stress (force/cross-sectional area) to account for potential differences in size between male and female?

RESPONSE: This is a great idea and topic for future work, but we feel it is beyond the scope of this study. To accomplish this goal, it would be necessary to characterize the three-dimensional arrangement of the various muscles in the complex, given that we measured bite force. This would be a formidable undertaking given the complexity of the musculature (see above response to comment on optimizing muscle length), joints involved in jaw adductor (paired on each side of head, potential cranial kinesis, etc. Sophisticated 3D computer modeling would probably be necessary.

COMMENT: Line 205 What do you mean by “tended”? Why not give a p value? Why conduct pairwise comparisons if the ANOVA shows no difference?

RESPONSE: You are absolutely correct. Thank you for pointing this out. We have removed any mention of the unfounded pairwise comparisons in the Results and Methods.

COMMENT: Lines 209-223 It would be helpful if you cited the separate figures in figure 2 individually in this results text to lead the reader through your findings. You also need to label the subsections of the figure A, B etc on the figure.

RESPONSE: We now cite separate parts of Figure 2 in the results text, and we have added labels to the figure panels and updated the figure legend.

COMMENT: Lines 239-240 Is it worth further clarifying that all jaw fibres tested reacted to all of these 3 antibodies suggesting a more homogenous fibre type in jaw than in thigh on the basis of the fibres you have tested, however recognising that there could be different fibre types untested in the muscle.

RESPONSE: We have reworded text in the Results and added to the Discussion. Thank you for the suggestion.

Discussion

COMMENT: Line 264: last stimulus to half tetanus relaxation in *Rana temporaria* extensor carpi radialis muscle was reported as 658ms for males in Navas and James 2007. Not sure why you haven't cited this value but have cited the paper elsewhere in your manuscript.

RESPONSE: Thank you for pointing that out. We have included the information in the Discussion with citation.

COMMENT: Line 326: or that another region of the jaw muscle you used contains fibres of other fibre type.

RESPONSE: You are absolutely right about this possibility, and we have added the suggestion.

COMMENT: Figure 3: have you got any better quality versions of these figures to be able to better discern the bands in the jaw muscle?

RESPONSE: Unfortunately, we do not. After working with multiple settings that is the best gel and image we could get. It was also only just discernible to the naked eye.

COMMENT: Figure 3 legend: under section B change reference to '4A' to '3A' I think we could do with some clearer explanation of what E, F and G represent.

RESPONSE: Corrected "4A" to "3A". Added clarification for E, F, and G images.